# Determination of Vitamin C in Foods Using the Iodine-Turbidimetric Method Combined with an Infrared Camera

**Yi Miao** [1], **Yuanyang Zhu** [1], **Wenzhu Zhao** [1], **Changshuai Jiao** [1], **Hongwei Mo** [1], **Xincheng Zhang** [1], **Sheng Liu** [1,*] and **Hongwen Gao** [2]

1   College of Computer Science and Technology, Huaibei Normal University, Huaibei 235000, China; MiaoYi321@outlook.com (Y.M.); 15556116568@139.com (Y.Z.); 15212626136@139.com (W.Z.); jiaocs@139.com (C.J.); mz324@139.com (H.M.); tediouscat@126.com (X.Z.)
2   College of Environmental Science and Engineering, Tongji University, Shanghai 200092, China; hwgao@tongji.edu.cn
*   Correspondence: liurise@139.com or Liusheng@chnu.edu.cn; Tel.: +86-183-6523-9378

**Abstract:** A novel method was proposed for the determination of vitamin C (VC) using an infrared camera combined with the iodine-turbidimetric method. Based on the redox between VC and iodine, the residual iodine was measured using the turbidimetric method with an infrared camera to obtain VC content. The light emitted by the infrared light-emitting diode (LED) was absorbed and scattered when it penetrated the residual iodine suspension. The transmitted light was captured by the infrared camera to form a digital image and the responding color components and grayscale values were obtained. The obtained color components and log-grayscale were fitted to the VC concentration, and the fitted relation expressions were used to measure the unknown VC solution. A VC measuring device equipped with an infrared camera and processing software was designed to obtain the color components corresponding to the images of the iodine suspensions. Compared with the spectrophotometry, the method based on the color component of brightness had a higher accuracy for measuring the VC standard solution. For VC measurements in tomatoes, nectarines, and VC tablets, our proposed method was highly consistent with spectrophotometry. Therefore, this method could potentially be implemented in the determination of VC in fruits and tablets, or other foods.

**Keywords:** food analysis; analytical method; vitamin C; iodine suspension; infrared camera; image processing

## 1. Introduction

Vitamin C (VC) is an essential micronutrient and appears as a colorless crystal that is soluble in water. VC is necessary for preventing scurvy and is often called ascorbic acid in medicine. VC plays an important role in the synthesis of neurotransmitters, steroid hormones, carnitine, the conversion of cholesterol to bile acids, tyrosine degradation, and metal ion metabolisms. The role of ascorbic acid as a biological reducing agent may be linked to its prevention of degenerative diseases, such as cancer and cardiovascular diseases. Thus, VC is of great importance to one's overall health. Most plants and animals synthesize ascorbic acid for their own requirements. However, humans cannot synthesize ascorbic acid due to a lack of the enzyme gulonolactone oxidase. Hence, ascorbic acid must be supplemented, mainly through fruits, vegetables, and tablets [1]. Therefore, an accurate measurement of VC in foods is important to determine VC intake and has a significant influence on dietary health.

There are many methods for the determination of VC, such as high-performance liquid chromatography (HPLC) [2–5], fluorescence spectroscopy [6], spectrophotometry [7], and electrochemical analysis [8–11]. In recent years, new methods have been proposed, including resonance Rayleigh scattering (RRS) [12] and flow injection photosensitized chemiluminescence [13]. However, these methods require expensive instruments, such as electron microscopes and photomultiplier tubes, or have complicated chemical reaction processes.

Spectrophotometry is used to determine VC based on the color reaction. By measuring the absorbance or luminescence intensity of light at a specific wavelength, the concentration of the product is obtained. It is often used for VC determination with a high sensitivity for 0–8 μg/mL, and was applied here to compare with our proposed method [7].

The development of computer image processing and imaging technology has made cameras widely used in various fields. In soil analysis, the RGB (red–green–blue) values of digital images and the derived soil indices were used to determine iron oxides and fines in soil [14]. A digital camera was also used for determining the iron and residual chlorine in water using N,N-diethylphenylenediamine [15]. In addition, image acquisition was carried out using a digital camera and image processing technology was used to acquire image information to determine the water turbidity, nitrite, ammonia nitrogen, sulfide, and phosphate contents, etc. [16,17]. The detection method that combines a digital camera and image processing technology has the advantages of simple design, convenient operation, and visualization, which verifies the feasibility of using a digital camera in trace analysis.

The turbidimetric method is often used for medical and biological detection [18,19]. The principle is that suspended particles absorb and scatter the incident light, and the intensity of scattering and absorption is positively proportional to the concentration of suspended particles. The concentration of the detected substances in the solution can be deduced by measuring the intensity of transmitted light. However, there is no turbidimetric method on the basis of iodine suspensions using a camera.

Most of the existing VC measuring methods require specialized instruments, or complicated operations, and cannot easily measure VC. There is a need for a VC measuring method that is easy to operate and does not require expensive instruments. The goal of the present work was to measure VC content using a typical infrared camera. In this work, an infrared camera was used to capture images of light penetrating through iodine suspensions that were prepared by the reaction between the aqueous VC solution and iodine ethanol solution. The VC concentration was obtained from the attenuation of incident light, which was indicated by the color component and gray information of the image.

## 2. Experimental Section

### 2.1. Measuring Device

The designed experimental device is as shown in Figure 1, and consists of a constant current driving circuit board for a light-emitting diode (LED), an infrared light source, a sample cell, and an infrared camera. The sampling area was sealed by a box that was made of light-excluding black acrylic, the thickness of which was 3 mm to isolate external light and, thus, avoid the effect of external light on the sample. The light source consisted of six 850 nm infrared LEDs, which were driven by a constant current circuit. Although the light from the LEDs was non-uniform, the measurement error was eliminated by calculating the average value of the whole imaging area. Compared with a single LED, the image sampling area was expanded, which reduced the impact of random errors, including the uneven distribution of suspended iodine and sampling area movements due to slight shakes of the device. When the infrared light passed through the suspension, it was absorbed and scattered by the suspended substance. Due to the different turbidity in the suspensions, the absorption and scattering varied and the image captured by the camera responded to the variation.

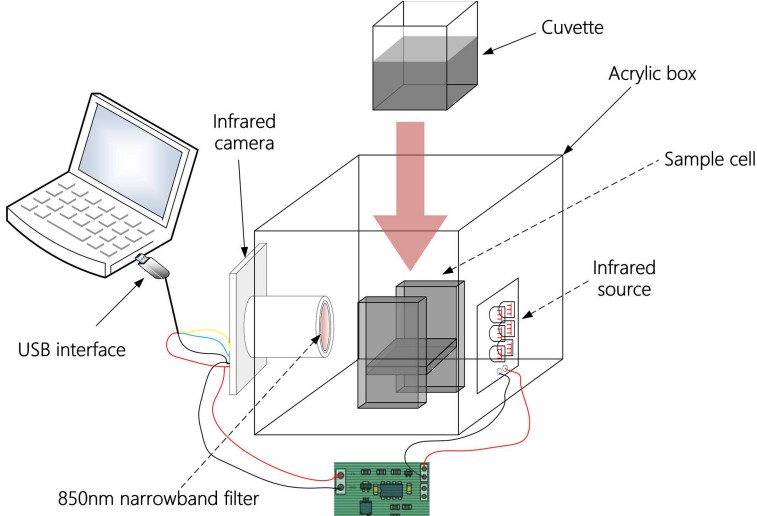

**Figure 1.** Structure diagram of the vitamin C (VC) measuring device based on an infrared camera.

### 2.1.1. Constant Current Circuit

The infrared LED was driven by a constant current circuit, as shown in the schematic diagram in Figure 2. U2 generated the reference voltage buffered by U1A, which was compared with the voltage on the sampling resistance R6. The comparison results controlled Q1 to realize the constant current driving of the LED. Adjusting W1 could change the working current of the LED. The circuit was powered by a universal serial bus (USB) port. After the circuit was powered, it required preheating for 20 min before the experiment could be carried out.

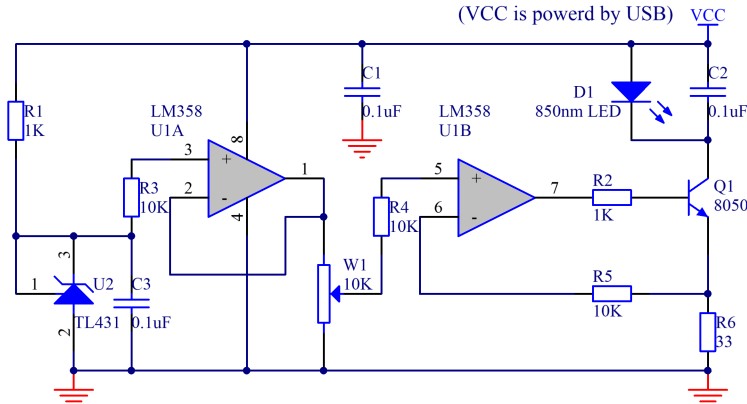

**Figure 2.** Light-emitting diode (LED) constant current driving circuit diagram.

### 2.1.2. Infrared Camera

The digital camera used for measurements is shown in Figure 3. It was a commercial ordinary infrared camera consisting of an optical lense, a CMOS (complementary metal oxide semiconductor) image sensor, and a signal processing circuit board. The camera model was KS1.3A142, produced by Shenzhen Kingsen Technology Co., Ltd. of China, and the maximum resolution of the camera was 1280 × 960 pixels. The lense was equipped with an 850 nm narrow-band filter, which could manually adjust the focal length to increase the clarity of the suspension image. The image acquired by the infrared camera was a single-channel image. These images are similar to grayscale images and avoid color interference from the test solution.

With the designed PC software, we could change the properties of the image by adjusting the camera parameters, such as the hue, saturation, and white balance, and ensure that the whole

measurement range had a good discrimination by adjusting the camera parameters of exposure, brightness, and contrast. These parameters of the camera were automatically saved to the registry of the system after setting. The same parameters should be used in the calibration and measurement process to ensure the accuracy of the measurement. Regarding camera selection, as long as the camera can manually change the focal length, disable the function of automatic exposure and brightness, and can adjust the exposure to ensure the consistency of the parameters during the measurement process, it can be used in the measurement system.

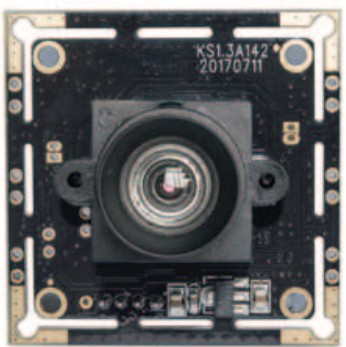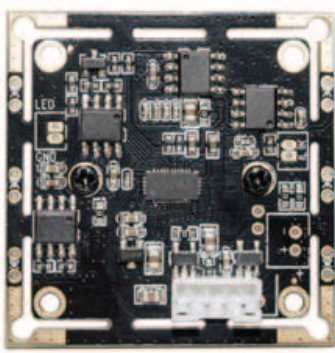

**Figure 3.** Camera printed circuit board (PCB) details.

The CMOS sensor used in the digital camera was a semiconductor component that was used to record light changes. Each pixel of the camera is equivalent to a photo detection element. When the light from the light source passed through the suspension, it was filtered by the 850 nm filter, and then projected onto the CMOS sensor. The signal processing component integrated in the camera processed the voltage data from the CMOS sensor to obtain the digital signal, and the digital signal was transmitted to the upper computer software through the USB port for further processing. The use of the digital camera replaces the photoelectric sensor, optical system, signal amplification, signal acquisition, analog-to-digital conversion circuit, and signal processing circuit that are required by other optical instruments. The design of the instrument is simplified, and measurement process can be visualized.

### 2.1.3. Software Design

The image acquisition software interface is shown in Figure S1 which was developed with C# and Camera_NET library in VS2012 platform. "Camera selection" was used to select the infrared camera. "Camera settings" were used to set the brightness, white balance, hue, saturation, and exposure of the camera. The software can automatically save the set parameters. "Snapshot the frame" obtained a frame image and took the average RGB value of the region of interest (ROI). In this way, the RGB value corresponding to the VC solution was obtained. Then, the corresponding grayscale and log-grayscale values were calculated and the corresponding Commission Internationale de L'Eclairage (CIE) Lab (CIE L*a*b*, L* for the lightness from black (0) to white (100), a* from green (-) to red (+), and b* from blue (-) to yellow (+)) values were obtained by converting the RGB color space to the CIE Lab color space. "Determine" opened the measurement part, which included zero calibration, span calibration, and measurement functions.

Grayscale used black tones to represent objects, and quantified the gray values from 0 to 255. "L" represents the brightness or the luminance of the light through suspension, "a" represents the range from red to green, and "b" represents the range from yellow to blue. L ranges from 0 to 100, and a and b range from +127 to −128.

Equation (1) was used to compute the grayscale and was calculated by a logarithmic grayscale:

$$grayscale = R \times 0.299 + G \times 0.587 + B \times 0.114. \tag{1}$$

An approximate conversion algorithm was used to transform the RGB color space to the CIE Lab color space as follows [20].

First, the RGB color space was converted to CIE XYZ color space (X, Y, and Z are extrapolations of RGB created mathematically to avoid negative numbers. Y means luminance, Z is somewhat equal to blue, and X is a mix of cone response curves chosen to be orthogonal to luminance and non-negative):

$$\begin{bmatrix} X \\ Y \\ Z \end{bmatrix} = \begin{bmatrix} 0.412453 & 0.357580 & 0.180423 \\ 0.213671 & 0.715160 & 0.072169 \\ 0.019334 & 0.119193 & 0.950227 \end{bmatrix} \begin{bmatrix} R \\ G \\ B \end{bmatrix} \tag{2}$$

$$\begin{cases} X &= \frac{X}{255 \times 0.950456} \\ Y &= \frac{Y}{255} \\ Z &= \frac{Z}{255 \times 1.088754} \end{cases}. \tag{3}$$

Then, the XYZ color space was converted to the CIE Lab color space,

$$\begin{cases} L &= 116 f(Y) - 16 \\ a &= 500[f(X) - f(Y)] \\ b &= 200[f(Y) - f(Z)] \end{cases} \tag{4}$$

$$f(t) = \begin{cases} t^{\frac{1}{3}} & if\ t > (\frac{6}{29})^3 \\ \frac{1}{3}(\frac{29}{6})^2 t + \frac{4}{29} & otherwise \end{cases}. \tag{5}$$

## 2.2. Chemical Reagents and Materials

In this design, sucrose, fructose, glucose, alumina, ammonium chloride, ferrous sulfate, copper sulfate, anhydrous calcium chloride, sodium chloride, ferric sulfate, magnesium chloride, vitamin C (ascorbic acid), vitamin B1, vitamin B2, L-methionine, L-tryptophan, L-lysine, L-cysteine, acetic acid, and kaolin were purchased from Sinopharm Chemical Reagent Co., Ltd. (https://www.reagent.com.cn, Shanghai, China). Vitamin E was purchased from Shanghai Ruichu Biotech Co., Ltd. All chemicals used were of analytical grade and used directly without further purification. The deionized water obtained from Shanghai Jingchun Water Technologies Co., Ltd. was used throughout the experiment.

Iodine ethanol solution: 0.7 g of iodine was added into 30 mL of 95% ethanol, with a FSH-2A adjustable high-speed homogenizer (Changzhou Guowang Instrument Manufacturing Co., Ltd., China; http://www.gwyq.net/) to accelerate dissolving. Then, the solution was transferred to a 50 mL volumetric flask and diluted to 50 mL with 95% ethanol.

VC standard solution: VC is stable in acidic solution but easily oxidized in alkaline and neutral solutions. Thus, 200 mg of VC was added to 200 mL of 0.25 M acetic acid to prepare 1 g/L of VC. Various VC standard solutions were prepared by diluting the 1 g/L of VC.

## 2.3. Sample Preparation

At 20 °C, 10 g of pant tissue of tomato or cucumber, or a crushed VC tablet, were put into a FSH-2A adjustable high-speed homogenizer. Then, 30 mL of 0.25 M acetic acid was added, and the mixture was quickly minced and homogenized with internal cutting for 4 min at 10,000 r/min into a homogenate. The homogenate was transferred to a 50 mL volumetric flask, and 20 mL of 0.25 M acetic acid added to dilute to the scale of 50 mL, before we shook well to obtain a sample solution. For a comparison of spectrophotometry, 0.5 g of kaolin was added to the sample solution, and then was

filtered to eliminate color interference. The filtrate was centrifuged for 6 min at 4000 r/min. The VC sample solutions were obtained.

*2.4. The Principle of Measuring Vc Using the Iodine-Turbidimetric Method*

The schematic presentation of the detection of VC is shown in Figure 4.

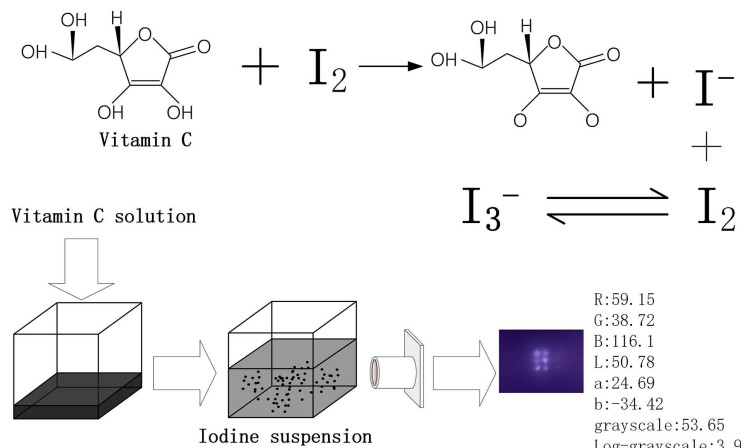

**Figure 4.** Schematic presentation of the detection of vitamin C (VC) with our proposed method.

When a VC solution is added to iodine ethanol solution, the solubility of iodine will decrease significantly due to the decrease of alcohol resulting from the VC solution, and an iodine suspension is formed [21]. Iodine is reduced to iodide ions and then iodide ions react with iodine to form triiodide, which is soluble. This reaction process is shown in Equations (6) and (7). The turbidity of the iodine suspension is relative to the VC content:

$$C_6H_8O_6 + I_2 = C_6H_6O_6 + 2\,H^+ + 2\,I^- \tag{6}$$
$$I_2 + I^- \rightleftharpoons I_3^-. \tag{7}$$

The 850 nm light source used in this study belongs to near-infrared (NIR, 780–1000 nm). NIR is widely used in turbidity measurements, and commercial turbidity instruments are based on 850 nm/860 nm light [22]. NIR light is less influenced by the color of the sample and is not sensitive to soluble particles, which essentially do not absorb and scatter infrared light, and the light band of the 850 nm light source is narrow ($\Delta\lambda = 50$ nm when I = 20 mA); thus, it is not absorbed by most organic substances [23]. Additionally, Maki et al. proved that $I_3^-$ has no sensitive light absorption in the NIR band [24], and J. G. Bayly et al. reported that the absorption of 850 nm light by water was small, and did not affect turbidity measurements [25]. Therefore, in the suspension, only suspended particles reduce the intensity of the transmitted light at 850 nm. With the exception of substances used to precipitate $I^-$ or with strong oxidation or reducibility, the soluble chemicals (such as pigments, inorganic salts, and sugars) do not interfere with the measurement [23], which is in keeping with our Interference Experiments. These results demonstrated the selectivity and feasibility of the method.

When light passes through a suspension, the suspended particles block the propagation of light in the suspension. To what degree the propagation of light is affected by the suspended substances depends on the size, shape, and composition of particles, and the wavelength of the incident light. In addition to the scattering effect, the transmitted light is absorbed by the particles and the light intensity reduces [23], and the change in the light intensity follows the Lambert–Beer law Equation (8) or Equation (9).

$$I = I_0 e^{-[\alpha a + \alpha b] x c} \tag{8}$$

$$\ln I = \ln I_0 - (\alpha a + \alpha b) x c \tag{9}$$

where $I$ is the intensity of the transmitted light, $I_0$ is the intensity of the incident light, $X$ is the length of the solution the light passes through, $c$ is the particle concentration, $\alpha a$ is the absorption coefficient, and $\alpha b$ is the scattering coefficient.

The suspensions with different turbidity were obtained by adding the VC solution to a certain amount of iodine ethanol solution. When light passes through the suspension, light is absorbed and scattered. The absorption and scattering affects the image, and this effect is mainly related to the turbidity of the suspension. After using the infrared camera to obtain the images of the incident light, the RGB values of the suspension were obtained by image processing. The grayscale and log-grayscale were calculated and the Lab values of the images were obtained by the color space conversion from RGB to Lab. By analyzing the relationship between the VC and RGB, and Lab and grayscale values, the relation expressions between the VC concentration and L, and grayscale and log-grayscale values were obtained, which were used to determine the VC in real samples.

*2.5. Statistical Methods*

The independent-sample T-test and one-way analysis of variance (ANOVA) were used with MATLAB2014 in this work to verify the reliability of the proposed method. After the independent-sample T-test between the standard concentrations and the data obtained by the proposed method, the P-value was obtained. The P-value—the level of significance—is the probability that hypothesis $H_0$ is true but refused according to the sample information. The test level $\alpha$ is usually taken as 0.05, and $P < \alpha$, $H_0$ was rejected, indicating that the measuring error of the proposed method should be considered. One-way ANOVA was used to analyze two data sets from two methods. For any given significance level $\alpha$, if the probability P-value was greater than $\alpha$, the original hypothesis should be accepted, that is there was no significant difference between the concentrations determined by the two measuring methods. In this study, $\alpha$ was 0.05.

*2.6. Measuring Procedure*

The aqueous VC solution (10 mL) was added to the iodine ethanol solution (1 mL) and mixed them uniformly to prepare the suspension. After transferring to a cuvette, the solution was placed into the measuring device and measured at room temperature. An image of the suspension was taken within 30 s, and the RGB and Lab values of the image were calculated automatically by the software. Before measurement, the measuring device was preheated for 20 min.

By measuring and fitting the Lab, RGB, grayscale, and log-grayscale values corresponding to a sequence of VC concentrations, the relation expressions were obtained. The L, grayscale, and log-grayscale that had higher goodness of fit and consistency were used in the actual sample's measurement.

## 3. Results and Discussion

The RGB values and Lab values of the suspensions were measured corresponding to different VC concentrations. A total of 16 standard aqueous VC samples prepared by diluting 1 g/L of VC solution using 0.25 M acetic acid were 0.5–5 µg/mL. All experiments were performed at a stable room temperature of 20 °C. Three groups of randomly selected experimental data were used to obtain a set of average data.

*3.1. Experiment Data and Analysis*

The color component data (R, G, B, L, a, and b), grayscale, and log-grayscale of the suspension image were obtained by the measuring device and analysis software. Figure S2a–c are the relationships between the R, G, and B values and the VC concentrations. The infrared images were similar to the gray images, and in theory, the RGB values are equal. However, the adjustment of camera parameters, including the white balance, hue, and saturation caused the observed difference between the R, G, and B values in Figure S2. The instability of the RGB to camera parameters is detrimental to the accuracy of the measurement. Thus, the RGB values were not suitable for the VC measurements.

Figure 5 and S3 show the relationship between the L, a, and b values and VC concentrations. Figure 5 shows an approximately linear relationship between L and the VC concentration in the range of 0–5 µg/mL, and a linear relationship is suitable for measurement. In addition, L is not disturbed by the camera parameters of white balance, hue, saturation, etc., because L represents lightness and is not affected by color. Therefore, the L value is a good choice for measuring the VC concentration. Figure S3a,b shows that different VC concentrations can correspond to the same a and b values in the range of 0–5 µg/mL. The relationship of the VC concentration to a and b cannot be expressed by simple functions therefore, the a and b values cannot be used to measure the VC concentration.

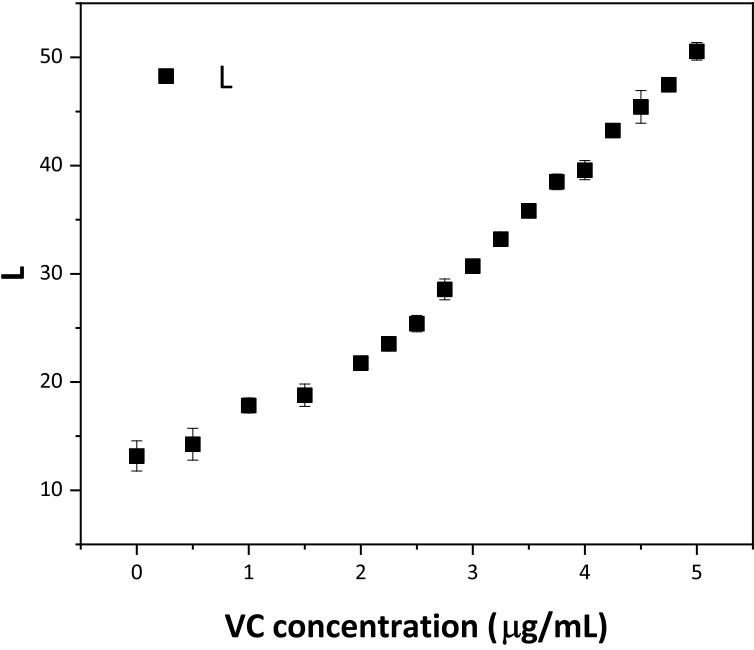

**Figure 5.** The relationship between the L value and VC concentrations.

Figure 6a,b show the relationships between the grayscale, and log-grayscale values with VC concentrations. Figure 6a shows that the grayscale value and VC concentration have a logarithmic relationship. In Figure 6b, the logarithmic grayscale and VC concentration are shown to be linearly related. The changes of grayscale and logarithmic grayscale conform to *I* and ln *I* in the Lambert–Beer law of Equations (8) and (9), therefore, grayscale is considered to linearly correlate to *I*. Therefore, grayscale and log-grayscale were used to measure the VC concentration.

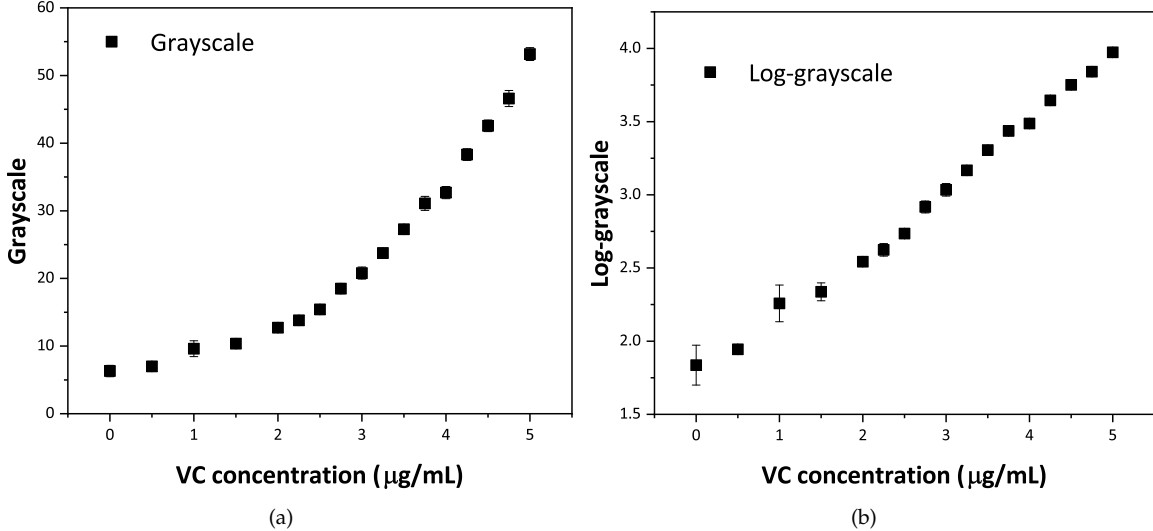

**Figure 6.** (**a**) The relationship between grayscale and VC concentrations. (**b**) The relationship between the ln(grayscale) and VC concentrations.

In Figures 6b and 5, the log-grayscale and L show a linear and approximately linear relationship to the VC concentration, and the comparison between L and log-grayscale after normalization is shown in Figure S4. According to Equations (6) and (7), the VC concentration is linearly related to the suspended iodine particle concentration. In Equation (8), there is a negative exponential relationship between the transmitted light intensity and the concentration of the suspended iodine particles. However, Figure S4 shows that L has a similar trend with the log-grayscale, linearly relating to the VC concentration. The reason for this is that L obtained by the imaging method is different from the light intensity obtained by the photocell of the turbidimeter. When L is converted from the RGB color space to the CIE Lab color space, the operation of Equation (2)–(5) is close to a logarithmic operation thus, the exponential relationship becomes approximately linear [23].

### 3.2. Fitting Results of Vc to L, Grayscale, and Log-Grayscale

The above experimental results show that the L, grayscale, and log-grayscale values have a high correlation to the VC concentration, and after fitting L, grayscale, and log-grayscale to the VC concentration, the fitting curves are shown in Figure 7. The adjusted R-Squared (Adj R-square), relation expressions, limit of detection (LODs), limit of quantification (LOQs), and ranges are shown in Table 1.

**Table 1.** The relation expressions, adjusted R-Squared (Adj R-square), relation expressions, limit of detection (LODs), limit of quantification (LOQs), and ranges.

| Figures of Merit | L | Grayscale | Log-Grayscale |
|---|---|---|---|
| Relation expression | $7.62x + 9.29$ | $e^{0.46x+1.66}$ | $0.44x + 1.75$ |
| Adj·$R^2$ | 0.98 | 0.99 | 0.99 |
| LOD (µg/mL) | 0.26 | 0.47 | 0.47 |
| LOQ (µg/mL) | 0.87 | 1.55 | 1.55 |
| Range (µg/mL) | 0.26–5.00 | 0.47–5.00 | 0.47–5.00 |

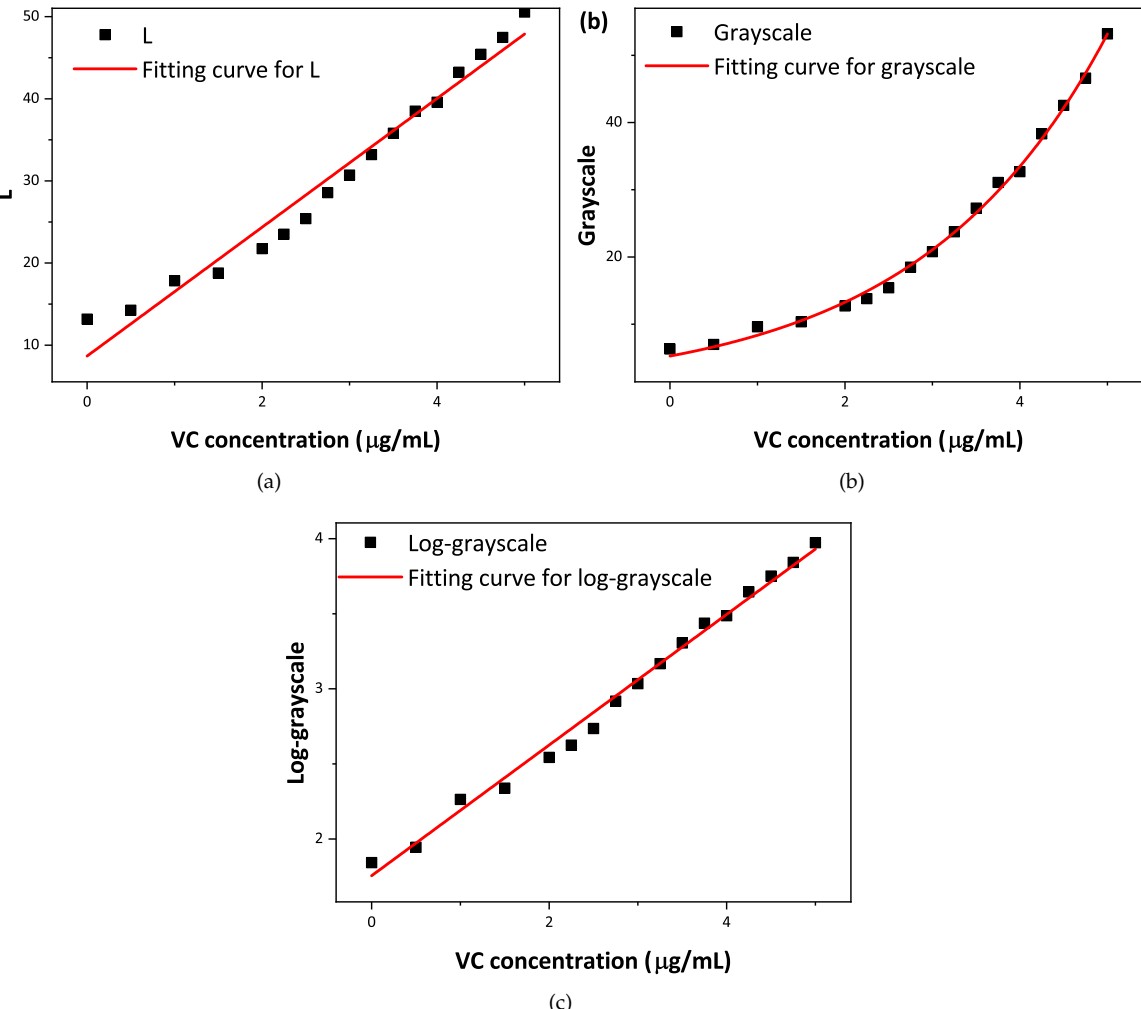

**Figure 7.** (**a**) The fitting curves of the VC concentration with the L value. (**b**) The fitting curves of the VC concentration with the grayscale value. (**c**) The fitting curves of the VC concentration with the log-grayscale value.

Adj R-square is a statistical indicator to reflect the degree of correlation between the variables. Adj R-square is calculated as a product-moment method, based on the same respective means of two variables based on the deviation, multiplied by two dispersions to reflect the degree of correlation between two variables. R-Square is the square of the correlation coefficient between the expression value (measured data) and the estimated value (calculated by the fitting model). Adj R-square is obtained by adjusting R-square according to the degree of freedom error. The closer to 1, the better the fit results. According to the fitting results, the L, grayscale, and log-grayscale values all had a high goodness of fit, with the log-grayscale value having the highest.

LOD, expressed as a concentration or quantity, is derived from the smallest measure that can be detected with reasonable certainty for a given analytical procedure. LOQ refers to the smallest concentration or the mass that can be quantitatively analyzed with reasonable reliability by a given procedure. According to International Union of Pure and Applied Chemistry (IUPAC) regulation, the LOD and LOQ were calculated as $LOD = 3SD/k$ and $LOQ = 10SD/k$ and were confirmed experimentally, where $SD$ is the standard deviation of the measured signal values of the blank samples (n = 20) and $k$ is the standard curve slope. In Table 1, the L, grayscale, and log-grayscale values all had a LOD lower than 0.5 µg/mL, and the L value had the lowest.

### 3.3. Interference Experiments

The influence of foreign substances, such as sugars, metal ions, amino acids, vitamins, and reductive acids was tested in the 4 µg/mL VC solution. The tolerance ratios of the maximum concentrations of the foreign substances that caused an approximately ±5% relative error in the VC concentration are listed in Table 2.

**Table 2.** Effects of foreign substances ([VC] = 4 µg / mL, n = 3).

| Foreign Substance | Tolerance Ratio | Interference Effect |
|---|---|---|
| $Na^+, Cl^-$ | 2000 | no significant effect |
| $NH_4^+, Cl^-$ | 1875 | no significant effect |
| $Cu^{2+}, SO_4^{2-}$ | 0.025 | $2Cu^{2+} + 4I^- = 2CuI \downarrow + I_2$ |
| $Fe^{3+}, SO_4^{2-}$ | 0.05 | $2Fe^{3+} + 2I^- = 2Fe^{2+} + I_2$ |
| $Al^{3+}, SO_4^{2-}$ | 1777.5 | no significant effect |
| $Ca^{2+}, Cl^-$ | 1550 | no significant effect |
| $Mg^{2+}, Cl^-$ | 1600 | no significant effect |
| $Fe^{2+}, SO_4^{2-}$ | 1725 | no significant effect |
| Cysteine | 0.05 | reducing $I_2$ |
| Methionine | 750 | no significant effect |
| Lysine | 0.075 | reducing $I_2$ |
| Tryptophan | 800 | no significant effect |
| Sucrose | 3000 | no significant effect |
| Glucose | 1000 | no significant effect |
| Fructose | 2000 | no significant effect |
| $Vitamin B_1$ | 300 | no significant effect |
| $Vitamin B_2$ | 200 | no significant effect |
| Vitamin E | Insoluble in water | no significant effect |
| Oxalic acid | 0.3 | reducing $I_2$ |
| Citric acid | 0.55 | reducing $I_2$ |

The interference experiment demonstrated that light metal ions, sugars, $Fe^{2+}$, and $NH_4^+$ could be allowed at very high concentrations, while some substances, such as those with reducibility and heavy metal ions, could be allowed only at low concentrations. However, the common plants generally contain little or no heavy metal ions, and thus do not affect the measurement. In addition, in the study combining the voltammetric technique with iodometry (of which the main chemical process was also the redox between $I_2$ and VC), Roxana A. Verdini et al. treated natural samples with ascorbate oxidase and carried out iodometric titration [26].

Compared with the average value of untreated samples (113 mg/100 g), the mean value of the treated samples was only 1.5 mg/100 g. This result proves that, although there are other reducing substances, the degree of reaction was very low, accounting for only 1.3% of the total reaction. Therefore, the interference of the reducing substances could be ignored.

### 3.4. Comparison with Spectrophotometry

Based on the above analysis results, we proposed a measurement method for measuring the VC concentration based on L, grayscale, and log-grayscale. In order to verify the accuracy and reliability of the method, the VC concentrations of standard solutions, fruits, and VC tablets were determined by the proposed method and compared with the measurement results by spectrophotometry. The spectrophotometry experiment used for contrast was based on the reduction of iron (III) by ascorbic acid to iron (II). Then the solution was complexed with 1,10-phenanthroline and the absorbance was measured at a 510 nm wavelength [7]. The spectrophotometer used for comparison was the 721G visible spectrophotometer (INESA Analytical Instrument Co., Ltd., China; https://www.instrument.com.cn).

3.4.1. Comparison Results for Standard Solution

The proposed method and spectrophotometry were used to determine the standard solutions of VC with 1, 1.5, 2, 2.5, 3, 3.5, and 4 µg/mL, respectively. After independent-sample T-testing of the VC concentrations, the P-value was obtained. The results show that the data from the measuring methods of L, grayscale, log-grayscale, and spectrophotometry were not significantly different from the standard VC concentrations. Table 3 shows the measured concentrations, P-values, and standard deviation (SD) of the four methods.

**Table 3.** The measured concentration values of L, grayscale, log-grayscale, and spectrophotometry on the standard solutions (n = 3), and the results of the independent-sample T-test and the standard deviation (SD) of L, gray, log-grayscale, and spectrophotometry.

| Measuring Method | Measured Concentrations (µg/mL) | | | | | | | P | SD |
|---|---|---|---|---|---|---|---|---|---|
| L | 1.02 | 1.25 | 1.92 | 2.24 | 2.88 | 3.47 | 4.01 | 0.062 | 0.15 |
| grayscale | 0.90 | 1.14 | 1.77 | 2.33 | 2.99 | 3.54 | 4.01 | 0.080 | 0.18 |
| log-grayscale | 0.90 | 1.14 | 1.76 | 2.34 | 2.99 | 3.54 | 4.01 | 0.079 | 0.18 |
| spectrophotometry | 1.04 | 1.43 | 2.07 | 2.53 | 2.92 | 3.70 | 4.34 | 0.224 | 0.16 |

In Table 3, the method L was closer to the standard VC concentration and had a higher accuracy. The methods of L, grayscale, and log-grayscale can be all used for the VC concentration measurement, and although gray and log-grayscale were less accurate than other methods, we suggest using gray or log-grayscale, as the transformation from RGB to Lab space was very time-consuming.

3.4.2. Comparison Results for Fruit and Tablet Samples

Three tomatoes and three nectarines selected from the market and a bottle of VC tablets, which can be purchased as an over-the-counter (OTC) drug (100 mg VC per tablet, made from Northeast Pharmaceutical Group Co., Ltd.), were used as test samples and treated to obtain the actual samples. The proposed methods and spectrophotometry were use to measure the samples and compare the VC concentrations measured by spectrophotometry with those of the proposed methods. The results are shown in Table 4. The VC concentrations in the VC tablets were measured as 90.325, 86.275, 85.850, and 85.108 mg per tablet (calculated from the concentrations of the sample solution in Table 4) by L, grayscale, log-grayscale, and spectrophotometry, respectively. These results can be regarded as normal for over-the-counter (OTC) drugs.

**Table 4.** Measurement of VC in fruits and tablets by the proposed methods and spectrophotometry (n = 3) and contrast results of L, grayscale, and log-grayscale, and spectrophotometry by one-way analysis of variance (ANOVA).

| Measuring Method | L (µg/mL) | | Grayscale (µg/mL) | | log-Grayscale (µg/mL) | | Spectropho-Tometry(µg/mL) | | P |
|---|---|---|---|---|---|---|---|---|---|
| | Mean | SD | Mean | SD | Mean | SD | Mean | SD | |
| Nectarine1 | 1.43 | 0.04 | 1.43 | 0.15 | 1.44 | 0.15 | 1.44 | 0.59 | 0.998 |
| Nectarine2 | 1.74 | 0.09 | 1.77 | 0.17 | 1.78 | 0.16 | 2.07 | 0.59 | 0.580 |
| Nectarine3 | 1.47 | 0.06 | 1.49 | 0.22 | 1.51 | 0.22 | 1.70 | 0.32 | 0.598 |
| Tomato1 | 3.11 | 0.14 | 3.20 | 0.04 | 3.20 | 0.04 | 3.33 | 0.56 | 0.838 |
| Tomato2 | 3.50 | 0.12 | 3.56 | 0.03 | 3.56 | 0.03 | 3.75 | 0.14 | 0.056 |
| Tomato3 | 4.57 | 0.01 | 4.49 | 0.04 | 4.48 | 0.04 | 4.49 | 0.60 | 0.982 |
| Tablet | 3.61 | 0.14 | 3.45 | 0.10 | 3.4 | 0.09 | 3.40 | 0.11 | 0.167 |

One-way ANOVA was used to respectively analyze the seven sets of data measured based on L, grayscale, and log-grayscale with the data of spectrophotometry. The probability P was also obtained. The P-value of one-way ANOVA for the seven sets of data is shown in Table 4, and all P-values were greater than 0.05, so there was no significant difference between L, grayscale, and log-grayscale and spectrophotometry, which verified the feasibility of the proposed method. In addition, recovery tests were performed with the standard addition method, the results of which are listed in Table 5.

**Table 5.** Analytical results of the recovery tests.

| No. | Method | VC Found (n = 5, μg/mL) | Relative Standard Deviation (RSD) (n = 5,%) | VC Added (n = 5, μg/mL) | VC Found after Adding (n = 5, μg/mL) | Recovery (n = 5,%) | RSD (n = 5,%) |
|---|---|---|---|---|---|---|---|
| 1 | L | 1.67 | 3.10 | 1 | 2.64 | 96.80 | 3.81 |
|  | grayscale | 1.66 | 6.05 | 1 | 2.69 | 102.42 | 4.60 |
|  | log-grayscale | 1.67 | 5.96 | 1 | 2.69 | 102.49 | 4.55 |
| 2 | L | 3.24 | 3.26 | 1 | 4.23 | 99.33 | 4.17 |
|  | grayscale | 3.25 | 1.85 | 1 | 4.26 | 101.13 | 3.08 |
|  | log-grayscale | 3.24 | 1.83 | 1 | 4.24 | 99.85 | 3.09 |

## 4. Conclusions

Our method using an infrared camera combined with the iodine-turbidimetric method was able to simplify the instrument design and replace the traditional optical instruments used in the VC analytics of fruits and VC tablets. This demonstrated the feasibility of a new method for VC measurements. By adding a VC solution to an iodine ethanol solution to form a suspension with the remaining iodine, the VC concentration was obtained by measuring the turbidity of the suspension with an infrared camera. The designed measuring device and image processing software were able to accurately acquire the corresponding color components of the suspension. The proposed method had high accuracy in standard solution measurements and no significant difference to the spectrophotometry in the VC measurements of the fruits and VC tablet, therefore, the method was proven to be effective.

Compared to the established VC measuring methods, the proposed method had the advantages of simple design and operation, visualization, as well as easy miniaturization and broadened the application scope of the image detection and turbidimetric methods. However, a disadvantage is that temperature had a great influence on the experimental results, which was found experimentally. This is because temperature can affect the solubility of iodine and, therefore, the reaction between iodine and VC. The method was not implemented on more tissue-dense foods. Therefore, following works should test the feasibility of this method on a wider range of foods and complete the compensation algorithm for different temperatures, which will provide more functionality to the method.

**Supplementary Materials:** The following are available online at http://www.mdpi.com/2076-3417/10/8/2655/s1.

**Author Contributions:** Conceptualization, Y.M. and S.L.; Formal analysis, Y.Z.; Funding acquisition, S.L.; Investigation, W.Z. and X.Z.; Methodology, Y.M.; Project administration, S.L.; Resources, Y.M. and C.J.; Software, Y.M.; Supervision, S.L.; Validation, H.M. and H.G.; Visualization, Y.M.; Writing—original draft, Y.M.; Writing—review and editing, H.G. All authors have read and agreed to the published version of the manuscript.

**Funding:** This work is partially supported by the National Natural Science Foundation of China (no. 61671434) and the Natural Science Fund for Colleges and Universities of Anhui Province (No. KJ2017ZD32).

**Conflicts of Interest:** The authors declare no conflict of interest.

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
