# Peer review of "Determination of Vitamin C in Foods Using the Iodine-Turbidimetric Method Combined with an Infrared Camera"

_applsci, doi:10.3390/app10082655_

Round 1

Reviewer 1 Report

This manuscript describes a IR-camera for determination of vitamin C in foods.  The engineering is clearly done well and the device obviously works for its intended purpose.  This manuscript however, is deficient in several areas related to the fundamentals of chemical analysis.   There are a number of undergraduate level mistakes in the data presentation.

  1. LOD and LOQ are reported to too many digits.   By its nature LOD has a 33% uncertainty and LOQ has a 10% uncertainty.  Please review and correct the significant digits on these determinations.
  2. For all reported results, if the uncertainties are in the 1-5% range, then 3-4 significant digits is too many.  Review and correct throughout. 
  3. Be consistent with significant digits throughout.  Both numbers in the ranges on the last line of Table 1 should be reported with correct significant digits.
  4. In discussing LOD and LOQ were these calculations confirmed experimentally?  Did you make a standard and run it at that level.
  5. The tolerance ratio is a very unusual way to discuss the effect of interferences.  Why calculate this then you can simply report the concentration level of the interference that caused a deviation in the result for the standard solution?
  6. In Figures 11 and 12, do not "connect the dots". 
  7. In Figures 11 and 12 why noy use the actual concentrations on the x-axis?
  8. I do not get the difference in the experiments between figures 11 and 12.  The data are clearly different.  Please explain in more detail.

I suggest that the authors more closely review literature on quantitative analysis in journals that specialize and analytical chemistry for examples of how to properly present quantitative data.

Reviewer 2 Report

The presented manuscript describes an alternative approach for analyzing vitamin C (VC) in foods. While the approach is an interesting idea, the manuscript cannot be recommended for publication in its current state. Major revisions are mandatory including possibly additional experiments.  

Language revisions are needed, preferably by a native English. Other comments are listed below which need to be addressed before the manuscript can be considered for publication:

General

  • Ensure you add all the spaces accordingly, e.g. between reference and text.
  • Ensure all abbreviations used in the manuscript are explained at their first use and subsequently being used consistently.
  • Please add an introduction to why such method is needed. What is your reasoning for developing such method? In your conclusion you need to add the advantages and disadvantages of your newly developed method compared to the established ones.

Abstract:

  • L2: Oxidation-reduction = redox.
  • L8: “A measuring device” is not sufficient. What is the measuring device?
  • L12ff: You state here that you have measured nectarine, tomato, and a tablet. This is hardly enough evidence to state that the VC content of all fruit and vegetables can be measured using this method. Given that tomato is in fact biologically a fruit, you have not measured a vegetable. Thus the statement as given here is not supported by your data. You may say that this method may be suitable for other fruits but without further evaluation it is not known.

Introduction:

  • Please shorten your introduction to the important details. This is not a review paper as you stated your intentions are original research.
  • VC is an essential micronutrient, which should be the first thing to mention rather than its physical appearance.
  • Whether or not VC is the simplest molecule within all vitamins is debatable. Nicotinamide, pyridoxal, choline come to mind. Please revise, it is an unnecessary statement.
  • L27: Not sure how VC is important to “diet health”, this is an example where a native English speaker will be able to translate the intention of the authors.
  • L30ff: This is quite lengthy and unnecessary. This is not a review paper there is no need to explain techniques in such great detail. That’s why you add references. E.g. there are many conditions described for HPLC to determine VC, et you are focusing on one for no apparent reason for the reader. A quick literature research will come up with a range of papers describing such analysis and they should be referenced rather than focusing on these presented details of one among many. Same applies to all techniques described.
  • L41: You have established RRS already in L40 as abbreviation why are you not using it?
  • L40ff: the whole paragraph and the following paragraphs are very long with too many details. See first comment under “Introduction”.
  • L73: Lab values. You added a lot of unneeded details about other techniques, yet you are not explaining “Lab” even though this an important part of your research.
  • L70ff: please shorten the last paragraphs about your research to a brief explanation what your aims are and how you approach it. Details about how you approached your research questions is part of the Methods and Results sections, not the Introduction.

Methods:

  • L93: acrylic is a nice description but does not allow a conclusion whether is a clear or light-excluding box. Please clarify.
  • L110: CMOS? Please see comment about abbreviations above.
  • L112: 1280*960 what? Please add units.
  • L112: “lense”.
  • L115: Who designed the software?
  • Figure 5 should be removed or moved to Supplemental materials. There is no added value to this figure for the reader. This is an article, not an SOP.
  • L163: See comment above, you have not analyzed a vegetable. You have analyzed nectarines and tomato, so please refer to this tissue as a general term of fruit and vegetables usually suggest a big range of different species, which is not the case here.
  • L164: model and manufacturer of the homogenizer?
  • L166: How much acetic acid did you add? Which scale? Please clarify.
  • L169ff: As far as I can tell the sample preparation of the fruit tissues and the tablet are the same, so please combine the sample preparation notes. Details like temperature, possible light exclusion, homogenizing conditions are missing. Prepare a separate sample preparation paragraph as this falls not under Chemicals and Materials.
  • First sentence under 2.3. should be omitted as the information provided is conclusive with the subsequent information provided. There are no line numbers for the first paragraph here.
  • L185: please add reference(s) for your statement.
  • L197: Please rename your paragraph. You are not describing a general procedure, which then could also include the sample preparation as it belongs to your procedures. Here you are referring to the actual measurement.
  • L198: It is an aqueous VC solution, right? How did you prepare the suspension? Mixing? Incubation? Conditions here need to be added.

Results and Discussion:

  • L208: Standard solutions in what solvent?
  • Figures 7-10: Please concise the graphs to the important ones. The remaining curves which have not been used for quantitation can be kept in supplemental materials. Again this is not a SOP.
  • L234: Not all your graphs in Figure 7 and 8 show a linear relationship. Only 8b is linear, all other ones are not linear. Please edit your statement to reflect the curves.
  • Table 2: Please add a column for interference effect (see L269) since there is plenty of space in your table. Please do not repeat values and information already presented in the table in the text, this is why you have a table.
  • In all your measurements there are no indications of replicates measured. Please add. If no replicates were done, you will have to repeat your experiments.
  • L285: Again, you did not measure a vegetable. You also avoided measuring a more tissue dense food item and stayed with foods with higher water content, which will not allow any conclusions of the suitability of your method for less watery food items. Will potato be a matrix suitable for this VC analysis?
  • Instead of explaining your statistics in a comparison results section you need to add a methods section for your statistics that explains all the e=tests used for this study.
  • 4.1 can be concised considerably, again, a native English speaker can help to simplify your convoluted text.
  • Figure 11 not useful in its current state. No replicates are indicated. It appears to be the measurement of a standard only. If so this can all be put in a table with numbers and statistics to show which ones of these measurements are sig. or not sig. different from the actual concentrations. Without the statistical we don’t actually know how well the measurement compare. Please change to a table with means, SD or CV, and indication of differences.
  • Table 4: again where are the replicates? Please indicate or repeat your experiment. Moreover, the concentrations indicated in the table are not matching the ones in the text. Please correct accordingly. difference need to be indicated.
  • Figure 12: is useless in current state. Figures and tables need to be readable and understandable as stand-a-alone. Sample serial number is not acceptable, what is what? There is no code description with the Figure. Again this should be table as explained above (Figure 11).

Reviewer 3 Report

This is innovative methodological research to simplify analysis for Vitamin C content with turbidity and camera photography. It cuts cost and promotes speed. Congratulations!

Author Response

Thank you very much to the reviewers for good comments and hard work.

Round 2

Reviewer 1 Report

The authors have sufficiently responded to the review comments. 

Author Response

(The authors gave the same response as above.)

Reviewer 2 Report

The authors have addressed my comments and concern in the revised manuscript. The manuscript could be recommended for publication after some additional edits:

  • L19: Please omit “essential nutrient”, you have already established that VC is an essential micronutrient in L18.
  • L53/54: The statement needs some backup. On what grounds do you establish to why such method is needed?
  • L53: “… that is easy to operate…”
  • L58/59: this is more Experimental design than Introduction. Usually, Introduction is used to explain the current state of knowledge, the missing pieces in this knowledge which then will lead to why your research is needed. Figure 1 and this sentence should be moves accordingly.
  • L149ff: I would rephrase to “At 20°C, 10g of pant tissue, or a crushed VC tablet, were put into….”. Thus, you are clearly separating that the plant tissue and the tablet are 2 different analyses.
  • L149: Cucumber? What happen to the nectarine? Please revise the plant tissues you were using and be consistent throughout the manuscript.
  • L193ff: please add the statistical software used for the analysis.
  • All your graphs should indicate error bars (e.g., SD, CI). For method development, you cannot just run one sample for each of your concentrations, you have to show the retreated measurements (not just technical replicates). The error bars indicate precision of your method.
  • Table 4: Are you referring to 3 replicates measured of the same fruit? Are the 3 nectarines and tomatoes 3 different fruits? Again, to show how well your method works you need to show replicate measurements of the same fruit, and present the mean and SD (or any other average concentrations with indication of variation). You cannot really combine different tomatoes or nectarines as each fruit needs to be seen as an individual, and they can all have different concentrations naturally. Thus, please indicate your averages and variation for each of the nectarines and tomatoes. For each calculated concentration. Then you can compare the results obtained by the different methods within the same fruit individual. Thus, your p-values should be shown for each row, not column.
  • Table 5: RSD is given for the recovery rates? Please add RSD for the VC found measurements.
